# Expert organisations with "challenging" and "complex" service users: Representation in English and Welsh autism charity reports and accounts

Helen Abnett[1], Kathryn Williams[2,3], Willow Holloway[3,4], Aimee Grant[5]*

1 School of Social Policy and Society, University of Birmingham, United Kingdom. Current affiliation: Centre for Research in Public Health and Community Care (CRIPACC), School of Health, Medicine and Life Sciences, University of Hertfordshire, United Kingdom, 2 School of Social Sciences, Cardiff University, United Kingdom, 3 Autistic U.K. CIC, Llandudno, Wales, United Kingdom, 4 Disability Wales, Cardiff, United Kingdom, 5 School of Health and Social Care, Swansea University, United Kingdom

* aimee.grant@swansea.ac.uk

## Abstract

The language and imagery used in Autism charities' communications influences societal understanding of, and attitudes towards, Autistic people. This, in turn, shapes perceptions of whether and how the disabling barriers in society experienced by Autistic people can and should be addressed. Yet, to date, there has been minimal exploration of this discourse employed by Autism charities. We, an Autistic-majority team of researchers, used critical reflexive thematic analysis to examine the language and imagery used in the Trustees' Annual Reports and Accounts of 11 large English and Welsh Autism charities. Representation within these reports emphasises adversities associated with Autism, and the language chosen to portray Autistic people largely describes Autism as an impairment. In contrast, charities represent themselves as the solution to the 'problem' of Autistic people, and thus deserving of increased resources and funding. Government is largely depicted as ineffective and deficient. We argue that these Autism charities are thereby upholding – rather than challenging – the disabling barriers in society experienced by Autistic people. For Autistic charities to better represent Autistic people and improve wider societal understanding of Autism, there is a need for more positive portrayals that challenge the 'charity saviour' trope in charity communications.

## Introduction

### Background and context

**Autism.** Autism is a lifelong neurodevelopmental difference. Current estimates, based on Autism prevalence in eight-year-old children in America, suggest that 3.2% of people may be Autistic [1] (as a marker of identity and culture, in this paper we

**Data availability statement:** These reports are currently available on the register of charities published by the Charity Commission for England and Wales. Links from which the documents can be downloaded are given below. Please note that the Charity Commission regularly updates this information to publish the most recent five years' worth of reports; therefore the reports are not available in perpetuity and links may not be stable. The National Autistic Society - https://register-of-charities. charitycommission.gov.uk/en/charity-search/-/ charity-details/269425/accounts-and-annu- al-returns?_uk_gov_ccew_onereg_chari- tydetails_web_portlet_CharityDetailsPortlet_ organisationNumber=269425 Autism Initiatives Group - https://register-of-charities. charitycommission.gov.uk/en/charity-search/-/ charity-details/5090836/accounts-and-an- nual-returns?_uk_gov_ccew_onereg_chari- tydetails_web_portlet_CharityDetailsPortlet_ organisationNumber=5090836 North East Autism Society - https://register-of-charities. charitycommission.gov.uk/en/charity-search/-/ charity-details/1028260/accounts-and-an- nual-returns?_uk_gov_ccew_onereg_chari- tydetails_web_portlet_CharityDetailsPortlet_ organisationNumber=1028260 Autism Together - https://register-of-charities. charitycommission.gov.uk/en/charity-search/-/ charity-details/1007878/accounts-and-an- nual-returns?_uk_gov_ccew_onereg_chari- tydetails_web_portlet_CharityDetailsPortlet_ organisationNumber=1007878 Ambitious About Autism - https://register-of-charities. charitycommission.gov.uk/en/charity-search/-/ charity-details/3940203/accounts-and-an- nual-returns?_uk_gov_ccew_onereg_chari- tydetails_web_portlet_CharityDetailsPortlet_ organisationNumber=3940203 Priors Court Foundation - https://register-of-charities. charitycommission.gov.uk/en/charity-search/-/ charity-details/3950583/accounts-and-an- nual-returns?_uk_gov_ccew_onereg_chari- tydetails_web_portlet_CharityDetailsPortlet_ organisationNumber=3950583 EASP - https:// register-of-charities.charitycommission.gov.uk/ en/charity-search/-/charity-details/1037868/ accounts-and-annual-returns?_uk_gov_ ccew_onereg_charitydetails_web_port- let_CharityDetailsPortlet_organisationNum- ber=1037868 Autism East Midlands - https:// register-of-charities.charitycommission.gov.

capitalise Autism and Autistic [2]). However, being Autistic can be intensely disabling in a world not designed for Autistic people. [3] Autistic people have differences in communication [4] and the processing of sensory information [2], although these may be masked or camouflaged to avoid negative responses from non-Autistic people [5]. Our previous analysis highlights that welfare policy in the UK often does not meet the needs of Autistic people, including Autistic people dying decades before their non-Autistic peers [6].

Historically, Autism was divided into groups using 'functioning' labels and differential diagnoses such as 'Aspergers syndrome'. This approach has been robustly critiqued [7,8], and a single diagnosis is now given to all Autistic people [9], with co-occurring diagnoses such as learning disabilities and support needs being noted separately [10]. Regardless of labels used, there is significant difference in lived experience, communication, sensory processing and support needs across the Autistic population.

**Models of disability.** Whilst not all Autistic people consider themselves disabled, models of disability can shed light on how Autism and Autistic people are perceived. Disability can be viewed through medical, charity, social, and biopsychosocial models. The medical model of disability focuses on the biological nature of impairments. In doing so, it situates the 'problem' of disability, and the onus on finding solutions, on the disabled person [11]. Furthermore, the medical model of disability has been associated with the paternalistic institutionalisation of disabled people, reducing their right to autonomy over their lives, including the social care that they receive. Similarly, in the charity model of disability, disabled people are seen as lacking agency, and deserving of pity [12]. Both the medical and charity models individualise disability, seeing it as a problem within its 'victims'.

By contrast, social models of disability highlight the role of disabling barriers in society. The social model thus aims for the full emancipation of all disabled people through removal of these barriers [13]. In the UK, disabled people led the adoption of a Strong Social Model of disability [11]. To date, however, this social model has not been consistently applied to Autistic people, focusing instead primarily on those with physical impairments [14]. Situated between the medical and social models is the biopsychosocial model of disability, combining both biological aspects of disability and psychosocial elements [15]. Within this paradigm, it has been argued that medical treatment should be patient centred, with the disabled person's needs and preferences prioritised [16].

**The role of charities.** In line with global trends [17], the voluntary, community and social enterprise (VCSE) sector is currently a crucial actor in social welfare service delivery in the UK [18]. Within the modern English and Welsh welfare state, it is often perceived that the role of charity is to fill gaps in provision to meet the needs of less-heard or 'underserved' groups, including Autistic people. Across the UK, thousands of Autistic people receive specialist care and education services from Autism charities, and these charities also have a key policy and influencing role. As examples, the National Autistic Society (NAS) acts as the secretariat for both the All-Party Parliamentary Group on Autism (APPGA) in the UK Houses of Parliament, and

uk/en/charity-search/-/charity-details/517954/
accounts-and-annual-returns?_uk_gov_
ccew_onereg_charitydetails_web_port-
let_CharityDetailsPortlet_organisationNum-
ber=517954 Autism Unlimited Limited - https://
register-of-charities.charitycommission.gov.uk/
en/charity-search/-/charity-details/1000792/
accounts-and-annual-returns?_uk_gov_
ccew_onereg_charitydetails_web_port-
let_CharityDetailsPortlet_organisation-
Number=1000792 Autism Anglia - https://
register-of-charities.charitycommission.gov.uk/
en/charity-search/-/charity-details/3940357/
accounts-and-annual-returns?_uk_gov_
ccew_onereg_charitydetails_web_port-
let_CharityDetailsPortlet_organisationNum-
ber=3940357 Autism at Kingwood - https://
register-of-charities.charitycommission.gov.uk/
en/charity-search/-/charity-details/1041924/
accounts-and-annual-returns?_uk_gov_
ccew_onereg_charitydetails_web_port-
let_CharityDetailsPortlet_organisationNum-
ber=1041924.

**Funding:** The author(s) received no specific funding for this work.

**Competing interests:** Kathryn Williams is a non-executive Research Director of Autistic UK CIC and Member of the Autism from Menstruation to Menopause Research Council Willow Holloway is: Director and Trustee of Disability Wales Director of Autistic UK CIC Trustee of Fair Treatment for the Women of Wales (FTWW) Trustee Rape and Sexual Assault Centre North Wales (RASAC) Member of The Welsh Government's Disability Task Force Member of The Welsh Government's Ministerial Advisory Group on Neurodivergence Member of The National Autism Teams Advisory Groups Co-Chair of North Wales Integrated Autism Service Strategy Board Member of the Autism from Menstruation to Menopause Research Council Aimee Grant is a non-executive Director of Disability Wales This does not alter our adherence to PLOS ONE policies on sharing data and materials.

the Senedd's Cross-Party Group on Autism; research conducted by these charities is regularly cited by government and parliament (see, for example, the House of Commons Library's 'Autism: Overview of policy and services' briefing) [19]; and the National Health Service (NHS) refers Autistic people and their carers looking for support to both national and local charities. Such roles contribute towards charities being accorded a privileged status, often positioned in policy discourse as the experts [20,21] best able to communicate the needs of those they seek to represent.

As Karaminis et al [22, p.3] demonstrate, "the language that society uses to understand autistic people both shapes and is shaped by the way autistic people are perceived and constructed". We similarly argue – as has previously been demonstrated regarding the press [22,23] and fictional media [24,25] – that Autism charities' public-facing communication "reflects, constructs and…reconstructs public attitudes and beliefs" around Autism [23, p.1093]. Both visual and text-based discourse have the power to affect how society views Autistic people; at its most extreme the linguistic separation of Autism from the Autistic person leading to "'altruistic' filicide" [26, p.872] (altruistic filicide refers to the idea of "'murder committed out of love'", which in cases involving disabled people can "dangerously overlap the idea of mercy killing") [27, p.47]. This separation also has the power to construct the idea of Autism being a separate entity to the Autistic person, a pathologizing fiction that upholds a medical model of disability in which Autistic people are considered as having deficits requiring treatment [28]. The language and imagery used to describe Autistic people is thus not a neutral act arising from preference. Rather, it has the power to shape how Autistic people are perceived, cared for, and problematised, particularly by people in positions of authority. This is relevant, as Autism is often stigmatised [29].

**Brief review of key literature.** In England and Wales, Autism charities therefore both directly impact Autistic people's capacity to flourish [30], and, through their communications, indirectly shape societal attitudes towards Autism. Charities "hold sway in their ability to…influence" [20, p.11], and charity discourses impact our understandings of society, and of each other. This discourse itself also reflects these charities' priorities, and the actions they are undertaking.

While there is a long-standing ethical debate regarding how charity 'beneficiaries' should be represented in charity fundraising and marketing materials [31], this discussion has primarily focused on the work of international development charities [32]. For example, Dhanani [33, p.28] explores the visual imagery used by ten large English and Welsh international development charities within their annual reports and annual reviews, and finds that these charities frequently present their "Southern constituents" as "dependent... passive, voiceless, and (culturally) backward". In contrast, the charities are portrayed as "reliable, knowledgeable and successful development experts" [33, p.28]. Similarly, in their analysis of the photographs used in the 2016 annual reports of eight of the largest US-based humanitarian organisations, Dhanani and Kennedy [34, p.351] find that the discourse used serves to undermine the "agency and, hence, humanity of beneficiary communities".

There has been limited application of this research approach to consider the representational practices of charities working with minoritised or marginalised groups

– including Autistic people – in countries such as the UK. This is despite the impact and influence of Autism charities in the UK as described above. The limited body of research which has studied Autism charities' representational practices has focused on these charities' adverts and websites. Such research has consistently found that these communications convey negative portrayals of Autism and Autistic people. For example, one study examined in detail the images and text in two Autism charity advertising campaigns in the UK and USA, and concluded the adverts provoke pity and fear, and remain embedded within the medical model of disability: one advert depicted Autism as "a child-enveloping monster that had to be destroyed to allow a boy to live a normal life" [35, p.219–220]. Three further studies consider the photographic and text-based representation of Autistic people on Autism charity websites, with one study from the USA finding that Autism is infantilised [36], and a replication study a decade later finding only limited change [37]. Analysis of the websites of South African disability charities finds that Autism is framed as "deficit, disability, disorder, disaster and disease" [38, p.12]. The present research seeks to add to this body of knowledge, by exploring and critically analysing the representational practice of large English and Welsh Autism charities.

## Materials and methods

**Aim:** To consider the representational language and imagery used within the Trustees' Annual Reports and Accounts of the largest Autism charities in England and Wales, using a Critical Autism Studies approach [39]. This paper has been written in accordance with the 2014 Standards for Reporting Qualitative Research (SRQR) checklist [40].

### Determining the study population

Our population of interest is English and Welsh charities whose primary stated objects are to support Autistic adults and/or children. Taking into account the perceived influence and trend setting role of larger charities [33], as well as the resources available to the research team, our analysis focuses on the largest charities.

   **Inclusion criteria.**

• Registered with the Charity Commission of England and Wales (CCEW)

• Income of £10,000,000 or more in their most recently reported financial year

• Solely providing support to Autistic adults, children, or both.

   **Exclusion criteria.**

• Subject to a statutory inquiry and/or under interim management.

   **Sample identification.**  Our inclusion criteria were operationalised as described below:

1. Search of the CCEW database:

Most English and Welsh charities with incomes of £5,000 or more must register with the CCEW. The CCEW maintains a searchable, public register of these charities, through which it publishes available Trustees' Annual Reports and Accounts for charities with incomes of £25,000 or more for up to the last five years. An 'Advanced Search' of the CCEW database was conducted to identify all currently registered charities; that included the term 'Autism' or 'Autistic' in their charity name, classifications, or objects and activities; that had reported total income of £10 million or more in their most recent financial year; and that were not subject to a statutory inquiry.

2. Manual screening of identified charities:

The websites and Trustees' Annual Report and Accounts of these charities were then screened by HA to include only those charities that self-identify as only providing support to Autistic adults and/or children.

## Data collection

**Data identification and extraction.** Each sampled charities' record on the CCEW database was searched to identify their most recent published Trustees' Annual Report and Accounts. The identified Trustees' Annual Report and Accounts was then downloaded directly from the CCEW database. Documents were available in pdf format.

## Data analysis

**Conceptual and theoretical approach.** Our analysis utilises Critical Autism Studies as a foundation [39]. We operationalise this by following Braun and Clarke's [41] six-phase reflexive thematic analysis, while paying critical attention to discourses. In practice, this means that our interpretation of the data was informed by Fairclough's [42] Critical Discourse Analysis, which provides the theoretical and conceptual bases for our approach [41]. Critical Discourse Analysis does not seek simply to describe such societal value systems, but to evaluate and explain them [42], and to suggest improvement. Therefore, following Terry and Braun [43], our analysis focuses on both semantic and latent features of the data. This enabled us to explore how organisations construct or position themselves and others within texts [44], so as to both draw attention to and challenge social inequities. Furthermore, the reflexive approach allowed us, as a majority Autistic team, to acknowledge our roles in the interpretation of the data rather than seeking objectivity [45]. As such, team discussions were group explorations and 'sense-checking' rather than consensus or calculating 'inter rater reliability'.

**Analytic process.** We describe our process below in line with Braun and Clarke's [41] six-phase approach to reflexive thematic analysis:

Phase 1 – Data familiarisation: This phase focused on both developing "deep and intimate" knowledge of the dataset, while also critically engaging with the Trustees' Annual Report and Accounts as data [41, p.42]. One author (HA) downloaded and shared the Trustees' Annual Reports and Accounts with the co-authors. HA actively read and re-read all 11 reports, while KW, AG and WH similarly familiarised themselves with five randomly selected Trustees' Annual Reports and Accounts, reviewing the other sources more briefly. The authors met on three occasions for discussion and reflection, and to generate initial codes.

Phase 2 – Coding: During these initial discussions, we identified three areas of potential critical analytic interest: the layout, format, and structure of the Trustees' Annual Report and Accounts; who is represented in the text and photographs of the reports, and how they are described; and whether and how individuals or groups are being asked to change their behaviour.

One author (HA) then conducted an in-depth coding process. This was an iterative process, involving systematic exploration of the documents, to code (potentially) relevant data. Each Trustees' Annual Report and Accounts was initially manually annotated, highlighting reference to different identities and groups within the reports. The reports were then uploaded to Nvivo 12, with the images and photographs used in each report extracted and uploaded separately with appropriate identification information. Data were coded by group or identity, and according to the broad codes developed during the data familiarisation process. Further, more detailed, codes were developed deductively, with the full corpus of Trustees' Annual Reports and Accounts re-reviewed as new codes were developed, until no additional codes were generated.

Phase 3 – Generating initial themes: These detailed findings were shared and discussed between the co-authors. During these discussions, the initial long list of codes was refined and clustered [41]. During these discussions, and in line with our theoretical framework as described above, our exploration of the data included a particular focus on understanding power. This further informed our consideration of whether the discourse "is asking for an individual or group to change their behaviour in any way" [46] which, drawing on Grant's [46] conceptualisation, we understand as 'calls to action'.

Phase 4 – Developing and renewing themes: This phase involves reviewing the viability of the initial thematic clusters, and developing the richness of the themes [41]. One author (HA) provided a detailed report of the data that had been clustered under each theme, which was then discussed and reviewed between the co-authors. This led to some re-working of the themes, including creating more distinct boundaries between the themes.

Phase 5 – Refining, defining and naming themes: during this phase, the authors collaboratively developed our theme definitions, testing for theme clarity, and refining the structuring and order of our themes. This includes centring the Autistic experience as the first theme discussed within this final paper.

Phase 6 – Writing up: Throughout phases 4, 5 and 6, our discussion and analysis were written up in a continuous process of reflection and feedback, drawing on Critical Autism Studies as our approach [39].

## Researcher positionality

KW, WH and AG are Autistic. HA is an experienced voluntary sector researcher, with experience of undertaking similar research [47,48], and a prior voluntary sector practitioner with involvement in writing Trustees' Annual Reports and Accounts. The remaining authors had not undertaken research on charities previously, although AG has undertaken discourse analysis in other areas focused on stigmatised groups [49]. KW, WH and AG are non-executive directors of disability organisations; KW and WH are non-executive directors of Autistic UK. AG and WH have experience of contributing towards the development of Trustees' Annual Reports and Accounts for a large disability charity.

## Results

This section begins with descriptive analysis of the characteristics of the charities and Trustees' Annual Reports and Accounts studied. We then present our analytical findings.

### Descriptive findings

**Characteristics of included charities.** We identified 11 charities meeting our inclusion criteria. As Table 1 shows, of the 11 identified charities, nine identify as working in social care, and eight provide education services. Four charities also describe themselves as engaging in advocacy and campaigning. The charities were also categorised based on their target demographic by age: seven charities stated they support those of all ages; one provided services only to adults; one states it supports adults and young people (aged 16+); and two work only with children and young people. The annual income of these charities ranges from £10 million to £96 million.

**Characteristics of the trustees' annual reports and accounts.** As shown in Table 2, there is substantial variation in the format and structure of these Trustees' Annual Reports and Accounts. Five Trustees' Annual Reports and Accounts are colour-documents, and six include images and/or photographs. The others are simple black and white documents with no photographs or images, and sometimes extremely limited description of the charity's activities. For example, of the 37 pages of North East Autism Society's Trustees' Annual Report and Accounts, only about two are devoted to the organisation's operational activities (others relate to management, governance, and financial information).

**Photographs.** Of the six Trustees' Annual Reports and Accounts studied that included photographs, there was variation in the quantity included. Whilst the Trustees' Annual Report and Accounts of NAS includes only three, the other five Trustees' Annual Reports and Accounts include between 22 and 54 photographs. 21 of these photographs were duplicates. Our dataset therefore included 151 unique photographs, of which 130 contain images of people. In most of the photographs containing people (n = 103; 79.2%), the people represented are not identified (by name, or as to whether they are staff, service users, volunteers, and so on). It is therefore not possible to concretely identify *who* is represented within most of these photographs.

### Analytical findings

These Autism charity Trustees' Annual Reports and Accounts represent several different actors, including Autistic people, their families, the charity, charity staff and trustees, central and local government, and other donors. This paper focuses on three of those groups: Autistic people, the charity, and government. These were chosen as the groups most relevant to

**Table 1. Charities included within selection (*N=11*).**

| Registered Charity Number | Charity Name | Recorded Income | Recorded Expenditure | Primary stated income source | Estimated proportion of income from government bodies[1] | Classification | | | Age group targeted |
|---|---|---|---|---|---|---|---|---|---|
| | | | | | | Social Care | Education | Advocacy & Campaigning | |
| 269425 | National Autistic Society | 95,633,000 | 94,835,000 | Fee income from statutory bodies | 82.1% | ✓ | ✓ | ✓ | All |
| 1170634 | Autism Initiatives Group | 71,181,000 | 69,654,000 | Local authorities | 88.0% | ✓ | ✓ | | All |
| 1028260 | North East Autism Society | 27,905,510 | 26,008,528 | Fee income | 98.9% | ✓ | ✓ | | All |
| 1007878 | Autism Together | 24,517,000 | 23,816,000 | Fee income, mostly local authority | 96.4% | ✓ | | ✓ | All |
| 1063184 | Ambitious About Autism | 24,473,000 | 23,423,000 | Local authorities | 78.3% | | ✓ | ✓ | Children and young people |
| 1070227 | Priors Court Foundation | 20,521,030 | 20,241,561 | Local authority | 99.3% | | ✓ | | Children and young people |
| 1037868 | Education and Services for People with Autism (ESPA) | 16,118,180 | 14,817,535 | Statutory bodies | 94.2% | ✓ | ✓ | | Adults and young people (16+) |
| 517954 | Autism East Midlands | 15,089,923 | 14,438,016 | Government funded bodies | 95.6% | ✓ | ✓ | | All |
| 1000792 | Autism Unlimited | 12,250,253 | 11,658,000 | Fee income | 99.2% | ✓ | ✓ | | All |
| 1063717 | Autism Anglia | 10,752,929 | 11,109,117 | Fee income | 99.4% | ✓ | | ✓ | All |
| 1041924 | Autism at Kingwood | 10,059,570 | 9,826,542 | Local authorities and NHS | 99.5% | ✓ | | | Adults |
| | | | | | Total: | 10 | 9 | 4 | |

[1]Estimates calculated based on the available data within the Trustees' Annual Reports and Accounts studied here. See Table 2 for further details.

**Table 2. Format and style of Trustees' Annual Reports and Accounts.**

| Organisation | Annual Report date | Length (pages) | Length (words) | Number of photographs | Report layout | Report style |
|---|---|---|---|---|---|---|
| National Autistic Society | 31/03/2022 | 75 | 25,998 | 3 | Portrait | Minimal colour; minimal photographs. |
| Autism Initiatives Group | 31/03/2022 | 56 | 21,788 | 0 | Portrait | Black and white; coloured logo but no other images |
| North East Autism Society | 30/04/2022 | 37 | 11,767 | 0 | Portrait | Black and white text; coloured logo but no other images. |
| Autism Together | 31/03/2022 | 46 | 16,419 | 0 | Portrait | Black and white; no logo or other images. |
| Ambitious about Autism | 31/08/2022 | 127 | 33,542 | 54 | Landscape | Highly presentation-friendly; numerous pictures, diagrams and stories. |
| Prior's Court Foundation | 31/08/2022 | 40 | 13,259 | 23 | Portrait | Highly polished and presentation-friendly; numerous photographs, colours, and logos. |
| ESPA | 31/03/2022 | 37 | 16,480 | 0 | Portrait | Black and white; no logo or other images |
| Autism East Midlands | 31/03/2022 | 52 | 14,921 | 25 | Landscape | Presentation-friendly; numerous pictures, images, and diagrams. |
| Autism Unlimited | 31/03/2022 | 56 | 14,021 | 45 | Landscape | Presentation-friendly; numerous pictures, images, and diagrams. |
| Autism Anglia | 31/08/2021 | 32 | 10,041 | 0 | Portrait | Black and white; no logo or other images. |
| Autism at Kingwood | 31/03/2022 | 56 | 15,238 | 22 | Landscape | Presentation-friendly; numerous pictures, images, and diagrams. |

our aim. Table 3 summarises our findings related to these three groups, while S1 Table provides a summary of our findings related to the other actors.

Next we describe how Autistic people, the charity itself, and government, are constructed within these Trustees' Annual Reports and Accounts. For each group, we focus first on how these Trustees' Annual Reports and Accounts describe that group, and then consider calls to action.

**Autistic people.** *Autistic people as impaired, infantilised, and othered:* In all 11 of these Trustees' Annual Reports and Accounts, most language choices made depict Autism as a deficiency, and of Autistic people as in some way impaired. This includes describing Autistic service users as having "challenging behaviours" (Autism East Midlands), or being "challenging learners" (ESPA), with "complex needs" (Autism at Kingwood). Autism itself is termed by both Autism East Midlands and the NAS as "autism spectrum **disorder**" [emphasis added], although the use of this term by the latter charity is buried deep in the notes of the accounts. This use of the word 'disorder' constructs Autism as an abnormality, with negative implications. Such impairment language is also reflected in charities' description of their use of 'Positive Behaviour Support' (PBS), which is specifically mentioned in the Trustees' Annual Report and Accounts of six of these charities, while two organisations refer to the "treatment" of Autism.

As shown in Table 3, in these Trustees' Annual Reports and Accounts, Autistic people are depicted as needy and infantile. A photo used by Autism at Kingwood, for example, shows the organisation's visiting founder using a white ceramic mug, while the adults she is visiting have colourful plastic cups. Such a difference suggests to a reader that Autistic people are either incapable of using ceramic crockery or have a child-like preference for colourful plastic.

In constructing the identity of Autistic people within these Trustees' Annual Reports and Accounts, the language chosen also "others" [50] Autistic people, both in relation to the charity, and the reader of the report. Autistic people are repeatedly described as 'them' or 'their' (not 'we' or 'our'). In five Trustees' Annual Reports and Accounts, this othering extends to Autistic people being commodified: ESPA, for example, suggest that employing Autistic people as software testers "recognises that many of the autistic characteristics can be a commercial asset in this field of work" (ESPA). Such language - 'the autistic characteristics' – homogenises and stereotypes Autistic people, while concurrently creating a discourse in which Autistic people are described as 'assets' to be exploited, rather than diverse individuals with their own agency.

**Table 3. Summary of findings.**

| Actor | Theme | Sub-theme | NAS | Autism Initiatives Group | North East Autism Society | Autism Together | Ambitious about Autism | Prior's Court Foundation | ESPA | Autism East Midlands | Autism Unlimited | Autism Anglia | Autism at Kingswood | Total |
|---|---|---|---|---|---|---|---|---|---|---|---|---|---|---|
| Representation | Autistic people — Impairment | Needy or challenging | ✓ | ✓ | ✓ | ✓ | ✓ | ✓ | ✓ | ✓ | ✓ | ✓ | ✓ | 11 |
| | | Autism as a condition/disorder | ✓ | ✓ | | ✓ | | | | ✓ | | ✓ | ✓ | 6 |
| | | Photographs depicting restraint | | | | | ✓ | ✓ | | | ✓ | | ✓ | 4 |
| | | Dehumanised/othered/infantilised | ✓ | ✓ | ✓ | ✓ | ✓ | ✓ | ✓ | ✓ | ✓ | ✓ | ✓ | 11 |
| | | Commodification | | ✓ | ✓ | ✓ | | | ✓ | ✓ | | ✓ | | 5 |
| | Voiceless/lacking agency | Made absent | ✓ | | ✓ | ✓ | ✓ | ✓ | | ✓ | ✓ | ✓ | ✓ | 8 |
| | | Spoken 'for' | ✓ | ✓ | ✓ | ✓ | ✓ | ✓ | ✓ | ✓ | | ✓ | ✓ | 8 |
| | | As sources of feedback | ✓ | ✓ | | ✓ | ✓ | | ✓ | ✓ | ✓ | | ✓ | 8 |
| | Tokenistic representation | As experts by experience | ✓ | | | | | | | ✓ | | | ✓ | 4 |
| | | Given voice by charity | ✓ | ✓ | ✓ | ✓ | ✓ | ✓ | ✓ | | ✓ | | ✓ | 8 |
| | Charity — Primary actor | As the subject | ✓ | ✓ | ✓ | ✓ | ✓ | ✓ | ✓ | ✓ | ✓ | ✓ | ✓ | 11 |
| | | As expert | ✓ | ✓ | ✓ | ✓ | ✓ | ✓ | ✓ | ✓ | ✓ | ✓ | ✓ | 11 |
| | | External quality markers | ✓ | ✓ | ✓ | ✓ | ✓ | ✓ | ✓ | ✓ | ✓ | | ✓ | 10 |
| | | Transformational reach or ambition | ✓ | ✓ | ✓ | ✓ | ✓ | ✓ | | | ✓ | | ✓ | 8 |
| | Facing challenge | Facing challenge | ✓ | ✓ | ✓ | ✓ | ✓ | ✓ | ✓ | ✓ | ✓ | ✓ | ✓ | 11 |
| | Government — Nature of partnership | As funder or fee-payer | ✓ | ✓ | ✓ | ✓ | ✓ | ✓ | ✓ | ✓ | ✓ | ✓ | ✓ | 11 |
| | | Collaborator | | ✓ | ✓ | | ✓ | | | ✓ | ✓ | ✓ | | 6 |
| | | Slow | | ✓ | | | ✓ | | | ✓ | | | | 3 |
| | Problematic | Under-funding | ✓ | ✓ | | ✓ | ✓ | ✓ | ✓ | ✓ | ✓ | ✓ | ✓ | 10 |
| | | Erratic or inconsistent | | | | ✓ | | | | ✓ | | ✓ | ✓ | 5 |
| | | Systemic or structural failures | | | | | ✓ | | | ✓ | | | ✓ | 3 |

*(Continued)*

Table 3. (Continued)

| Actor | Theme | Sub-theme | NAS | Autism Initiatives Group | North East Autism Society | Autism Together | Ambitious about Autism | Prior's Court Found-ation | ESPA | Autism East Midlands | Autism Un-limited | Autism Anglia | Autism at King-wood | Total |
|---|---|---|---|---|---|---|---|---|---|---|---|---|---|---|
| Autistic people | Change behaviour | Become more confident, communicative, or resilient | ✓ | ✓ | ✓ | | ✓ | ✓ | | ✓ | | ✓ | | 7 |
| | | To fulfil potential | ✓ | ✓ | ✓ | ✓ | ✓ | ✓ | ✓ | ✓ | ✓ | | | 9 |
| | | Use of PBS | | | | ✓ | ✓ | | ✓ | ✓ | ✓ | | ✓ | 6 |
| | Become less 'burdensome' | Become more independent | ✓ | ✓ | | ✓ | | ✓ | ✓ | ✓ | ✓ | | ✓ | 8 |
| | | Become employed | ✓ | ✓ | ✓ | ✓ | ✓ | ✓ | ✓ | ✓ | | | ✓ | 9 |
| Charity | Increase size | Expand reach | ✓ | ✓ | ✓ | ✓ | ✓ | ✓ | ✓ | ✓ | ✓ | ✓ | ✓ | 11 |
| | | Increase funding | ✓ | ✓ | ✓ | ✓ | ✓ | ✓ | ✓ | | ✓ | ✓ | ✓ | 10 |
| | | Raise profile | ✓ | ✓ | | ✓ | ✓ | ✓ | ✓ | | | | | 6 |
| | Raise profile | Influence policy | ✓ | ✓ | | | ✓ | | | | | | | 3 |
| | | Advocate/campaign | ✓ | | | ✓ | ✓ | | | | ✓ | | ✓ | 5 |
| | Develop capacity | Review and improve services | ✓ | ✓ | | ✓ | ✓ | ✓ | ✓ | ✓ | ✓ | ✓ | ✓ | 10 |
| | | Develop capacity | ✓ | ✓ | ✓ | ✓ | ✓ | ✓ | ✓ | ✓ | ✓ | ✓ | | 10 |
| | | Increase service user voice | ✓ | ✓ | | | ✓ | | | ✓ | | | | 4 |
| Government | Improve funding | Increase funding | ✓ | | | | ✓ | | | ✓ | | ✓ | ✓ | 5 |
| | Improve practice | Address lack of inclusion | ✓ | | | | ✓ | | | ✓ | ✓ | | ✓ | 5 |
| | | Improve education | ✓ | | | | ✓ | | | | | | | 2 |
| | | Address workforce issues | | | ✓ | | ✓ | | ✓ | ✓ | | | ✓ | 5 |

*Autistic people as lacking independent voice:* The portrayals in these Trustees' Annual Reports and Accounts also suggest Autistic people lack independent voice and agency. The direct absenting of Autistic people in these Trustees' Annual Reports and Accounts is demonstrated in the lack of explicit Autistic service user representation within the photographs used. Whilst it is not common in reflexive thematic analysis to quantify, we found it striking that in only 34 (26.2% of unique photographs containing people) are the people pictured clearly identified, and in only nine (6.9%) of these cases are these individuals identified as being service users. Other people identified are presented as trustees, staff members, donors, or ambassadors of the charity, and are not described as Autistic.

Further, Autistic people's parents, families, or other stakeholders, are often presented as speaking for, and instead of, Autistic people: Autism Initiatives Group describes how their "greatest pride is in compliments from families of the people we support", and lists compliments from three groups of actors - "From a family member", "From a social worker" and "From our staff" – but not Autistic service users themselves. Such a construction may suggest to the reader that Autistic people are either incapable of independent voice, or that such voice is less important than those of their families or carers.

Where Autistic people's voice is conveyed, this often appears tokenistic. Autistic people are presented as being involved, consulted, or associated with the work of the charity, but are not leaders or decision-makers. For example, Ambitious About Autism's Trustees' Annual Report and Accounts includes extensive first-person reports from Autistic young people. Yet, the framing of these accounts suggests that these Autistic people's voice is in fact contingent on the actions of the charity itself: the charity states "we involve pupils and learners in decisions", and "we invest in skills and resources to enable their participation in a variety of ways". The focus is on how the charity allows and enables Autistic young people to have voice. This presentation – superficially of Autistic voice – instead constructs the message that Autistic people *lack* intrinsic, independent voice, and instead are *given* this voice by the charity.

*The call to action: Autistic people need to change their behaviour:* The presentations of Autistic people within the Trustees' Annual Reports and Accounts thus creates a narrative that Autistic people should change their behaviour. As demonstrated in Table 3, this discourse is constructed across all 11 Trustees' Annual Reports and Accounts. In these constructions, Autistic people are shown as needing to become more confident, communicative, or resilient; to adapt so as to be able to fulfil their potential; and to reduce their challenging behaviour. Such a call to action is included even within those three Trustees' Annual Reports and Accounts (those of Ambitious about Autism, Autism East Midlands, and Autism at Kingwood) that also acknowledge that it is systemic inequity – rather than individual behaviour – which disables Autistic people.

This need for behaviour change is also associated with a stated need for Autistic service users to become 'independent' and/or to seek employment. 10 of these Trustees' Annual Reports and Accounts include such a call to action. Two Trustees' Annual Reports and Accounts explicitly link this to a need for Autistic service users to become less of a burden on the state, for example to "reduce their reliance on state benefits" (ESPA). Arguably, Autistic people, in this construction, are a burden; and the solution to this burden is for Autistic people to change.

**The charity.** *The charity as the expert:* Across all 11 of these Trustees' Annual Reports and Accounts, a key identity constructed of the charity is as an expert. This capability covers several different domains, including service provision, fundraising, financial management, COVID-19 protection and campaigning. Ten Trustees' Annual Reports and Accounts reinforce this description of expertise with the inclusion of external quality markers, describing positive inspections from the Office for Standards in Education, Children's Services and Skills (Ofsted) and Care Quality Commission (CQC), as well as a range of other awards and accreditations, such as the "Pride of Reading Inspiration Award" (Autism at Kingwood); and an "Excellence in Inclusion award from the East Midlands Chamber of Commerce" (Autism East Midlands). As suggested in the introduction, such a construction can contribute to a discourse in which charities are seen as having the authority to act for and speak on behalf of Autistic people. It is the charity's staff – rather than Autistic people themselves – that are perceived has having the greater understanding of Autistic lives.

*The charity as subject to pressure:* While presenting themselves as expert, in all 11 Trustees' Annual Reports and Accounts these charities also describe the challenges they face. Notably, these problems are largely portrayed as external

to the actions of the charities themselves. Pressures are described as being the result of the COVID-19 pandemic, and a wider environment of financial and staffing issues. Ambitious about Autism, for example, describes how both the pandemic and inflation rises "mean that [staff] recruitment and retention have become a challenge for us, like they have for many organisations", while Autism East Midlands similarly states that after the pandemic "along with most other employers, we find ourselves facing ongoing staff shortages". While these charities' positive traits (expertise and achievements) are therefore described due to their own capability, the challenges they face are presented as sector-wide, and due to circumstances beyond their control.

*The call to action: development and growth:* While Autistic people are thus presented as needing to change, the representation of the charity as expert is directly linked to the overwhelming message given throughout these reports that these charities need to grow and develop; to expand their services, profile, and reach; and to increase their income. Autism Initiatives Group's plans for example include "Further development of supported living services", while ESPA aims "to continue to develop new and innovative services". The NAS seeks to "increase the profile of [their] schools" while Autism Together's focus is on raising "autism awareness, acceptance and promot[ing] autism good practice". The primary message communicated by this language is that these charities need to do more, to be bigger and often better-known, and that they need more funding to enable them to achieve this.

**Government, both central and local.** *'Government' as the source of external challenge:* As shown in Table 1, our estimates based on available data suggest that all 11 charities included within this analysis are reliant on statutory (government) income. This financial relationship is recognised explicitly in all these Trustees' Annual Reports and Accounts, with central and/or local government described as funder or fee payer in all Trustees' Annual Reports and Accounts studied.

However, while recognising government as their key funders, across almost all these Trustees' Annual Reports and Accounts, central and/or local government are also subject to repeated critique. The exception is the North East Autism Society, in which government is solely presented as a funder and partner. The negative portrayals constructed include that local and/or central government: provides inadequate funding; is inefficient and/or erratic; and are slow to act. Autism at Kingwood describe how local authority funding increases "appear arbitrary, and therefore not based on true calculations", while Autism Anglia similarly notes the "inconsistency of fee income increases" from local authorities, and the NAS emphasises the challenges posed by "continuing local authority budget constraints". These language choices portray a government that is anomalous and unreliable.

Similarly, Ambitious about Autism shares the story of 'Arran' (name changed in the report), of which excerpts are included below [emphasis added]:

> The learner family support team found a suitable college placement for Arran, **but the local authority refused to fund it. They insisted that Arran should settle** for a [home-based social care package]…[After challenges from the charity, the local authority] did offer to send in an educational psychologist to assess Arran…but **there were concerns over the inaccuracy and independence of the report provided by the psychologist**. The learner family support team **challenged the parts of the report that were inaccurate**...The local authority **finally agreed** to fund Arran's placement at college.

As shown, this representation directly juxtaposes the charity's represented capability with suggested incompetence by the local authority: the authority is described as 'insisting' on an 'inappropriate' placement. The identity suggested by this text is of a local authority that is both obstinate and flawed. In juxtaposition, it is those working for the charity that are expert and efficient. Arran himself – and his family – are largely voiceless as actors within this account.

*The call to action: increased funding:* As described above, within these Trustees' Annual Reports and Accounts most charities describe under-funding from government as a key issue. However, in their calls to action only around half overtly

call for increased funding from government. This includes Ambitious about Autism, which emphasises the need for a transparent and efficient funding system for SEND. Autism East Midlands similarly asks for a "long term funding solution", and Autism Anglia requests "government bodies pay appropriate fees for contracted services". This is supported by a range of other calls for action, including addressing challenges within the SEND education system; improving social care workforce recruitment; and addressing wider issues within statutorily provided services. Overall, as Table 3 shows, calls to action from government are less present in these Trustees' Annual Reports and Accounts than calls for Autistic people to change, or for charities to grow.

## Discussion

In this paper, we aim to critically analyse the discourse of Autism charity Trustees' Annual Reports and Accounts. Overall, we argue that throughout these Trustees' Annual Reports and Accounts, Autistic people are largely portrayed as 'problems', as needy, challenging, and a burden. This is accompanied by language that suggests that Autistic people need to change, to become less 'burdensome'. The charities themselves are represented as expert, with a concomitant focus on an apparent need for growth. Government is primarily portrayed as a barrier to the effective provision of services for Autistic people by the Autism charities studied, due to inadequate funding for these charities, as well as ineffective and inconsistent policy approaches.

The narrative of Autistic people as needy, challenging, and a burden, lacking in voice and capability, is in stark contrast to difference-based narratives presented by Autistic adults, [51] including those who are non-speaking, [52] and fails to adequately account for known deficiencies in public services when used by Autistic people in the UK – including, for example, a lack of healthcare clinician understanding of Autistic presentations of pain and distress, and unequal access to education, employment, and housing services. [6] As such, we find the presentation of service users within these Autism charity Trustees' Annual Reports and Accounts is similar to the presentation of international development charities' 'beneficiaries' by charities based in England and Wales [33] and the USA, [34] in which these charities' constituents are depicted as "reliant, dependent, passive, backwards, voiceless, unintelligent and needing care". [33] These Trustees' Annual Reports and Accounts also reflect comparable findings to that found in the prior research – described above – studying Autism charity advertising in the UK and USA, [35] and websites in the USA [37] and South Africa, [38] which similarly found negative portrayals of Autism, with Autism being portrayed as a deficit and disorder. The othering of Autistic people found in this paper also reflects findings from prior critical studies, in which the professionalisation and marketisation of charity organisational practices is critiqued as leading to the 'beneficiaries' of charities being made "different". [50]

The language and photographs used within these Trustees' Annual Reports and Accounts arguably construct a demeaning discourse about Autistic people. It is the 'impairment' of Autistic people that creates challenges, rather than socially constructed barriers. Such a language construction, we argue, has implications for wider societal attitudes and behaviours towards Autism and Autistic people. This is demonstrated even within these Trustees' Annual Reports and Accounts themselves, in that six charities describe their activities as involving PBS. Applied Behaviour Analysis (ABA) – from which PBS is derived – has weak evidence of effectiveness [53], and there is some evidence it is associated with post-traumatic stress disorder (PTSD) in Autistic adults. [54] PBS actively seeks to make Autistic people behave in a non-Autistic way, and the acceptance of such approaches is allowed – we argue – by the construction of discourse such as we have found here that portrays Autism as an abnormality.

In direct juxtaposition to this negative presentation of Autistic people, within these Trustees' Annual Reports and Accounts, all these charities present themselves as expert. As Dhanani [33] found in her study of international development organisations, it is the charity that is portrayed as "superior and able", in contrast to the 'pitiful' Autistic person. Our findings also align with other prior research, [47] which found that international development charities were utilising such narratives to demonstrate to their (potential) donors their deservingness of increased funding. We argue that these charities' focus on a need for growth is key to the construction of the discourse we find here: that Autistic people lack agency

and are in need of 'help', and that charities need to grow to address these 'problems'. These discourses distort understanding of the systemic barriers Autistic people face.

The relationship between charities and Autistic people as represented within these Trustees' Annual Reports and Accounts is therefore one in which these charities portray themselves as having agency and capability, whilst Autistic service users (and other Autistic people) are othered and voiceless. The relationship presented with government is perhaps more dualistic: despite all these charities relying on government funding, and almost all these charities critiquing government, only around half explicitly argue that government funding needs to be increased and/or funding systems improved. This may suggest a power imbalance, with charities not seeking to explicitly criticise government, on which they rely for funding. Yet, we argue that while these charities do not explicitly call for an increase in funding, the construction of government created by this discourse is of central and local government as being inefficient, inexpert, and erratic. This again constructs a discourse in which, as the 'expert', it is solely the charity that has the agency to act towards and for Autistic people.

Our findings therefore demonstrate that the visual and text-based discourse constructed by these charities is situated within the medical and charity models of disability – both of which individualise disability, and frame disabled people as 'victims', lacking in agency. As Shakespeare [11] demonstrated, responsibility for the 'problem' of Autism is placed on Autistic people. In contrast, the charity is portrayed as active and expert, and as "knowing what is good for" [55] Autistic people. The social model of disability, based on a rights-based approach that seek to address societal barriers to the emancipation of disabled people, is not reflected in this discourse.

Further, in this paper we demonstrate the value of applying a lens to UK charities which has hitherto largely focused on the field of international development charities. We find similar themes here that have been more widely demonstrated within the international development discourse, but have not previously been robustly evidenced within the discourse of English and Welsh charities working with minoritised or marginalised groups – including Autistic people – in countries such as the UK. Our findings also show how the discourse constructed by the Autism charities studied here reflects the "non-disabled saviour" trope that has been found to be common in popular culture [56], but has again not thus far been studied extensively within other discourses. As such, we believe this paper contributes to a deeper understanding of the construction and operationalisation of the medical and charity models of disability in contemporaneous English and Welsh charity discourse.

## Strengths and limitations

The contributions of this study are three-fold: we add to the empirical literature by providing detailed consideration of the representational practices of 11 of the largest English and Welsh Autism charities; conceptually, we demonstrate the value of applying a lens to domestic charities which has hitherto largely focused on the field of international development charities; and practically, we seek to provide detailed and specific demonstration of the discourse constructed by these Autism charities, thereby helping practitioners to consider and reflect on their own organisational discourse.

This study is limited by our focus on the Trustees' Annual Reports and Accounts of a small number of high-income charities. The use of Trustees' Annual Reports and Accounts allows us to explore in-depth a rich stream of visual, text, and accounting data. However, we do not consider the wider communications activities of these organisations. Nor do we have a clear understanding of exactly how these Trustees' Annual Reports and Accounts are developed. Prior research suggests that the primary audience for Trustees' Annual Report and Accounts is a charities' donors or funders, including government funders, [47] and the presentation of the discourse in these documents is likely to influence the nature of the discourse found. Further research would be beneficial in considering communications targeted at a broader, or different, audience, and particularly in understanding the extent of meaningful leadership and participation of Autistic people in the construction of these Trustees' Annual Report and Accounts. In addition, the limited information regarding the photographs included in these Trustees' Annual Reports and Accounts restricted our analysis of this visual data. Research focused specifically on interpreting visual information shared by Autism charities through different formats would again add value

to our understanding of Autism charity discourse. Finally, our focus on only the largest Autism charities means our findings are not generalisable beyond the specific charities included in this study. Nevertheless, we argue that this research provides important insight for how Autistic people are portrayed in large Autism charity discourse.

## Conclusion

This paper explores the discourses constructed in the Trustees' Annual Reports and Accounts of 11 of the largest English and Welsh Autism charities. Our findings identified that Autistic people are constructed as impaired, needing to change, and lacking agency. As noted above, this is in line with both the medical and charity models of disability. Autistic people are also presented as deserving of pity, and lacking voice and agency. The social model of disability, based on a recognition of the rights of Autistic people and the need to remove disabling systemic barriers in society – including in regards to Autistic people's unequal access to health, education, employment, and housing services [6]- is not reflected in the discourse constructed by these Trustees' Annual Reports and Accounts. We find that Autistic people's voices are being excluded from these discourses, in favour of those of their families or carers, or the charity themselves. We therefore call for Autism charities to carefully reflect on the discourses that are constructed through their communications. These discourses matter, because of both the operational and representational power and influence held by charities. One way of addressing the challenges we found may be to ensure meaningful and accessible Autistic representation as leaders within these charities. We also call for further academic research that applies the lens we use here to the discourse of other actors, within and beyond the disability sector, when discussing Autism.

## Supporting information

**S1 Table. Findings related to actors other than Autistic people, the charity, and government.**
(DOCX)

## Author contributions

**Conceptualization:** Helen Abnett, Kathryn Williams, Willow Holloway, Aimee Grant.

**Data curation:** Helen Abnett.

**Formal analysis:** Helen Abnett, Kathryn Williams, Willow Holloway, Aimee Grant.

**Methodology:** Helen Abnett, Kathryn Williams, Willow Holloway, Aimee Grant.

**Supervision:** Aimee Grant.

**Writing – original draft:** Helen Abnett.

**Writing – review & editing:** Helen Abnett, Kathryn Williams, Willow Holloway, Aimee Grant.

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
