## [Decision Letter · Decision Letter 0]

24 Jun 2025

PONE-D-25-27468

Expert organisations with “challenging” and “complex” service users: representation in English and Welsh Autism charity reports and accounts

PLOS ONE

Dear Dr. Grant,

Thank you for submitting your manuscript to PLOS ONE. After careful consideration, we feel that it has merit but does not fully meet PLOS ONE’s publication criteria as it currently stands. Therefore, we invite you to submit a revised version of the manuscript that addresses the points raised during the review process.

We look forward to receiving your revised manuscript.

Kind regards,

Michal Soffer

Academic Editor

PLOS ONE

Journal Requirements:

Kathryn Williams is a non-executive Research Director of Autistic UK CIC and Member of the Autism from Menstruation to Menopause Research Council.

Willow Holloway is:

Director and Trustee of Disability Wales

Director of Autistic UK CIC

Trustee of Fair Treatment for the Women of Wales (FTWW)

Trustee Rape and Sexual Assault Centre North Wales (RASAC)

Member of The Welsh Government's Disability Task Force

Member of The Welsh Government's Ministerial Advisory Group on Neurodivergence

Member of The National Autism Teams Advisory Groups

Co-Chair of North Wales Integrated Autism Service Strategy Board

Member of the Autism from Menstruation to Menopause Research Council

Aimee Grant is a non-executive Director of Disability Wales

Reviewers' comments:

Reviewer's Responses to Questions

**Comments to the Author**

1. Is the manuscript technically sound, and do the data support the conclusions?

Reviewer #1: Yes

Reviewer #2: Yes

Reviewer #3: Yes

2. Has the statistical analysis been performed appropriately and rigorously?

Reviewer #1: N/A

Reviewer #2: N/A

Reviewer #3: Yes

3. Have the authors made all data underlying the findings in their manuscript fully available?

Reviewer #1: Yes

Reviewer #2: No

Reviewer #3: Yes

4. Is the manuscript presented in an intelligible fashion and written in standard English?

Reviewer #1: Yes

Reviewer #2: Yes

Reviewer #3: Yes

5. Review Comments to the Author

Reviewer #1: This is a well constructed and implemented study and makes an important contribution to an under-researched field. The authors have presented the data clearly and the paper is well written and carefully argued. I look forward to seeing it published.

Reviewer #2: Responses to review questions

Question 1: Is the manuscript technically sound, and do the data support the conclusions?

Partly (but mostly yes), the study draws on data extracted from the Charity Commission of England and Wales. The methodology is robust (distinct inclusion and exclusion criteria and detailed analysis) and the data analysis supports the conclusions. One thing missing was the year of the reports. The methods state that the reports were from the last five years, but no dates are provided in the text or the tables. The authors use the acronym “TARA”, however, the meaning is not provided. The interpretations of the visual analysis was less convincing since 6 of 11 reports had photos and in the majority of the photos it was not possible to identify who was represented within.

Question 3: Have the authors made all data underlying the findings in their manuscript fully available?

No, I viewed the link provided and it was not connected directly to the data (i.e., the TARA reports used for the analysis). I am uncertain how to access to the data on the website provided.

Question 4: Is the manuscript presented in an intelligible fashion and written in standard English?

Yes, however there are a few typos:

Pg. 4 The last sentence has a direct quote – page number needed

Pg. 6 The last sentence ends with a comma and is incomplete. I also think that it is odd to state the findings this soon in the manuscript.

Pg. 7 Part of the aim is to focus on the imagery, however, there was no background on how autism (or disability more broadly) imagery has been analyzed in the past. The background only focused on language/discourse.

Pg. 8 TARA is used but no definition is provided – it is in the abstract but never written out with (TARA) in the text.

Pg. 11 Typo – centering (not centring)

The page numbering of the manuscript starts at page 1 again after the tables.

Pg. 8 – what is INGO?

Pg. 9 (new page numbering) Second paragraph has a couple of direct quotes – page number needed.

Throughout the manuscript Disability is capitalized. Can the authors provide a note on why this is the case? Furthermore, Autism (not sure why?) and Autistic (which makes sense) are capitalized. Please justify the use of capitalizations.

Overall: The manuscript, titled “Expert organisations with “challenging” and “complex” service users: representation in English and Welsh Autism charity reports and accounts” offers a critical reflexive thematic analysis of language and imagery used in Trustees’ Annual Reports and Accounts of 11 large English and Welsh Autism charities. I appreciate the inclusion of autistic researchers in this analysis and the critical and reflexive approach taken. The findings are not at all surprising given the construction of autism within the medical model of disability. There are a few points I would like the authors to consider in the revision of this manuscript.

1. In the introduction of the manuscript, it would be important for the authors to have a statement about how the differences of experiences by autistic people in communication, masking, sensory information processing, etc. vary across a spectrum. The heterogeneity of autism is an important aspect to articulate and will help situate the study.

2. I appreciate the clear articulation of the medical, charity, social and psychosocial model of disability. It was unclear if the authors were taking a social or psychosocial model approach in their analysis. There is alignment with findings to the medical model but not the other models – not even the charity model. How does this research add to our understanding of these models of disability?

3. Relatedly, although the author’s cite Yergeau’s book, I am surprised that the study is not framed using Critical Autism Studies approach, which aligns with much of the analysis: challenges medicalization and deficit model, autistics as agents and producers, examines how institutions like the charities under investigation shape the understanding and lives of autistic people, reflexive approach, etc.

4. Pg. 5 – Can the authors elaborate on what is meant by “altruistic’ filicide”?

5. Pg. 6 – in the brief review of literature, the authors may also want to add that charity discourses reflect the priorities of the charities and the work that they do, what they fund, the programs they create/sponsor, etc. thus the discourses reflect the actions they are taking.

6. Pg. 6. The manuscripts states that there is an “impact and influence of Autism charities in the UK”. Given this statement, the manuscript should provide some information about the influence/impact of autism charities in the UK. What percentage of funding (compared to direct billing by NHS and other state/government programs) is from charities? Do charities cover care only (as opposed to research?)? Some contexts here is needed to show how these charities really matter when it comes to financial investments (and the discourses they carry about autistic people).

7. Although the photos were unequally represented in the reports and most did not identify who was in the photo (autistic, care giver, etc.), I am curious if age and gender were also coded, especially given the focus on childhood autism and over representation in boys. In the US, race and ethnicity are also underrepresented in the media/culture/discourses. This is one area that I found missing even in the discursive analysis.

8. Pg. 1 (new page numbering) – in the analytic findings, the authors mention “non-human actors” – I am curious what is meant by this phrase since all the actors listed are human entities (even charities). Machines, technologies, diagnostic tools, policies, etc. would be non-human, but this is not what was analyzed. One way to consider this is that the reports themselves serve as a non-human actor given the discursive work that they do beyond the human production of the reports? I guess the use of non-human in this sentences seems out of place.

9. Pg. 8 – Discussion – this is related to the previous comment about the UK context – the authors state “Government is primarily portrayed as a barrier to the effective……..” My question as a reader – why is this important in the context of the UK, which has National Health Care? Knowing more about the types of national service programs dedicated to autism through NHS or other governmental programs would be helpful. Furthermore, when stating “known deficiencies in public services when used by Autistic people” – are the authors referring to services outside of the charities and if so, can they provide an example?

10. Pg. 9 – Discussion - I appreciate the critique of PBS, However, there are cases where PBS can be helpful for autistic people, especially if they are at risk of injuring themselves. This is extreme, of course, but must also be acknowledged. The authors critique to lack of heterogeneous representation of autism in the TARA and should equally be reflexive of the heterogeneity of different approaches that autistic people have benefited from to have greater voice and representation. There is a fine line that the authors should acknowledge.

11. Pg. 10 – Discussion – I would have liked to learn more about the systemic barriers that Autistic people face? How does this analysis support these barriers?

12. As stated before, these findings are not that surprising. To help add to the significance of these findings, can the authors say something about how autism is different than other disabilities as it relates to the findings? Or is this just another example of how differences are viewed as disabling in medical models of disability? In other words, how might the analysis of autism bring new insight into the pathologization of disability? Charity model of disability? Critical Autism Studies?

13. Limitations – one question that came up is the target audience of these reports? Who is the target audience and how might this shape the discourse? Are they meant for broad general audience? Government funders? Autistic people and their families? Clinical professionals?

Reviewer #3: This study is an important contribution that I recommend for publication. Its focus on the messaging around support of autistic individuals is not only timely but critical, addressing an area often overlooked and insufficiently explored in current literature. The authors effectively highlight systemic issues in the charity system that has intentions of "help" but in turn are causing harm.

Beyond this human rights dimension, this work holds immense importance for the autistic community itself. It offers a vital voice, recognizing and validating the experiences of autistic individuals, and how important language and framing are for the community.

6. PLOS authors have the option to publish the peer review history of their article (what does this mean? ). If published, this will include your full peer review and any attached files.

**Do you want your identity to be public for this peer review?**  For information about this choice, including consent withdrawal, please see our Privacy Policy .

Reviewer #1: No

Reviewer #2: No

Reviewer #3: No

---

## [Author Response · Author response to Decision Letter 1]

30 Jul 2025

Please see attached document, for a detailed description of our revisions.

Many thanks,

Aimee and Helen on behalf of the authors

---

## [Decision Letter · Decision Letter 1]

7 Oct 2025

Expert organisations with “challenging” and “complex” service users: representation in English and Welsh Autism charity reports and accounts

PONE-D-25-27468R1

Dear Dr. Grant,

We’re pleased to inform you that your manuscript has been judged scientifically suitable for publication and will be formally accepted for publication once it meets all outstanding technical requirements.

Kind regards,

Ramandeep Kaur

Academic Editor

PLOS ONE

Additional Editor Comments (optional):

Reviewers' comments:

Reviewer's Responses to Questions

**Comments to the Author**

Reviewer #1: All comments have been addressed

Reviewer #3: All comments have been addressed

2. Is the manuscript technically sound, and do the data support the conclusions?

Reviewer #1: Yes

Reviewer #3: Yes

3. Has the statistical analysis been performed appropriately and rigorously?

Reviewer #1: N/A

Reviewer #3: N/A

4. Have the authors made all data underlying the findings in their manuscript fully available?

Reviewer #1: Yes

Reviewer #3: Yes

5. Is the manuscript presented in an intelligible fashion and written in standard English?

Reviewer #1: (No Response)

Reviewer #3: Yes

Reviewer #1: The authors have responded to the reviewer comments and made appropriate changes. The paper is of merit and interest to autism researchers and disability service providers and I look forward to seeing it published.

Reviewer #3: This manuscript is an essential and timely contribution to the field. The authors have diligently and effectively addressed all of the previous reviewer comments, strengthening the paper. The revisions have made an already important work even more robust and compelling. I recommend its acceptance for publication.

**Do you want your identity to be public for this peer review?** For information about this choice, including consent withdrawal, please see our Privacy Policy

Reviewer #1: No

Reviewer #3: No

---

## [Editor Report · Acceptance letter]

PONE-D-25-27468R1

PLOS ONE

Dear Dr. Grant,

I'm pleased to inform you that your manuscript has been deemed suitable for publication in PLOS ONE. Congratulations! Your manuscript is now being handed over to our production team.

Kind regards,

on behalf of

Dr. Ramandeep Kaur

Academic Editor

PLOS ONE